# A Novel Cost Calculation Method for Manipulator Trajectory Planning

**DOI:** 10.3390/s24134096

**Published:** 2024-06-24

**Authors:** Leiyang Fu, Shaowen Li

**Affiliations:** 1School of Information and Artificial Intelligence, Anhui Agricultural University, Hefei 230036, China; fly2008@ahau.edu.cn; 2Anhui Provincial Key Laboratory of Smart Agricultural Technology and Equipment, Hefei 230036, China

**Keywords:** manipulator, trajectory planning, cost calculation, UR5, inverse kinematics

## Abstract

It is worthwhile to calculate the execution cost of a manipulator for selecting a planning algorithm to generate trajectories, especially for an agricultural robot. Although there are various off-the-shelf trajectory planning methods, such as pursuing the shortest stroke or the smallest time cost, they often do not consider factors synthetically. This paper uses the state-of-the-art Python version of the Robotics Toolbox for manipulator trajectory planning instead of the traditional D–H method. We propose a cost function with mass, iteration, and residual to assess the effort of a manipulator. We realized three inverse kinematics methods (NR, GN, and LM with variants) and verified our cost function’s feasibility and effectiveness. Furthermore, we compared it with state-of-the-art methods such as Double A* and MoveIt. Results show that our method is valid and stable. Moreover, we applied LM (Chan λ = 0.1) in mobile operation on our agricultural robot platform.

## 1. Introduction

Autonomous operation is an essential goal for the development of robots, and trajectory planning determines the path and intensity of a manipulator’s movements [1]. Trajectory planning represents a structural sequence of a manipulator at each moment to achieve specified pose requirements. Each structural sequence is related to the states of each joint, such as position, angle, and speed. Good trajectory planning reduces the distance between points (origin and destination), minimizing execution time and the amount of work required by a robot [2].

Trajectory planning involves the kinematics and dynamics of a manipulator. Kinematics studies the relationship between displacement, speed, and acceleration of the movements of each joint, and dynamics studies the relationship between joint motion and force. From the kinematic view, trajectory planning can be performed in rectangular and joint coordinate spaces [3], and the planned trajectory must be continuous and smooth. Trajectory planning in a Cartesian coordinate space refers to expressing an end-effector’s posture, velocity, and acceleration as a function of time. Reversely solving the displacement, velocity, and acceleration values of each joint is based on the information of the end-effector. The Cartesian coordinate space is beneficial to humans because it is relatively intuitive. Trajectory planning in a joint coordinate space refers to expressing each joint variable of a manipulator as a function of time and solving the state of the end-effector from the forward pose of each joint. The joint coordinate space needs to be more intuitive for humans, but the combined output of joints has no problems with multiple solutions. Trajectory planning usually needs to meet some constraints, such as path optimization, smooth motion, obstacle avoidance, and optimal energy consumption. 

When considering operating efficiency and safety, the relationship between joint speed and force, that is, dynamic issues, is crucial in trajectory planning. It includes the forward problem of calculating joint motion with known forces and the inverse problem of estimating the required forces based on the joint speed corresponding to the known trajectory. Inverse kinematics is the problem of determining the corresponding joint coordinates given the position and attitude of an end-effector. There are two methods for solving inverse kinematics: (a) the analytical method and (b) the numerical method. The analytical method refers to mathematical formulas and derivation methods, such as the Jacobian matrix method, which converts the robot’s kinematic model into a system of mathematical equations. The angles of each robot joint are obtained by solving the system of equations. Analytical methods usually cannot optimize additional conditions (such as joint limits). Sometimes, the solution may not exist. Numerical inverse kinematics uses iterative techniques and can consider additional constraints such as collision avoidance, joint limit, and maneuverability [4].

Manipulator trajectory planning algorithms can be divided into two categories: algorithms based on motion relationships and algorithms based on heuristic searches. A trajectory planning algorithm based on motion relationships uses known information, such as initial position or historical posture, to summarize the manipulator’s trajectory planning pattern. The motion of the robotic arm is expressed as a nonlinear equation. The limits of joint angle, velocity, or acceleration are transformed into constraints, and the trajectory planning problem is transformed into a nonlinear optimization problem. Potter et al. [5] proposed an energy-saving optimal trajectory planning algorithm for redundancy manipulators based on quadratic polynomials. They proposed an optimal weighting vector to determine the effect of each joint on total energy consumption. Pei et al. [6] divided the trajectories into several sub-trajectories. They used an inverse kinematics-based model to obtain the joint angular displacement corresponding to the final position of the end-effector in each subsector. To ensure the stability of the movement, the fifth polynomial curve was used to interpolate the continuous angular process in the joint space. In general, trajectory planning algorithms based on motion relations can quickly find solutions that meet the constraints. Still, they have the disadvantages of only finding local optimal solutions, requiring convergent gradients, and not satisfying discontinuous functions. A trajectory planning algorithm based on a heuristic search usually proposes the criterion for evaluating the optimization goal. Then, it applies the heuristic search method to find the optimal global solution. The heuristic search performs an orderly search of the planning domain so that the generated trajectory is close to the optimal trajectory. A variety of constraints can be used, and compatibility is vital. Agarwal [7] proposed a trajectory planning method based on a new fuzzy clustering model and established an expert system by obtaining link attitude data. With the help of an expert system, it is possible to obtain optimal solutions with various objectives. However, this approach relies too much on the richness of the data. Lin [8] proposed a hierarchical genetic algorithm for manipulator path planning consisting of a global path planner and a local motion planner. The international path planner plans the path for the end-effector of the robotic arm, and the local motion planner uses a non-random initial population genetic algorithm to plan the configuration of the robotic arm along the path. The planned trajectory is smoother, but the convergence is slower. In short, a trajectory planning algorithm based on a heuristic search can find the optimal solution under specific evaluation criteria. Still, the calculation time is long, and it is not suitable for real-time trajectory planning.

## 2. Background

Existing trajectory planning methods are practical, but a particular scene, such as a manipulator used in intelligent agriculture, needs the flexibility to respond autonomously. The background of this paper is on how a manipulator can operate dexterously (grabbing, cutting, and spraying) in agriculture. For example, a robot for field work uses battery power, and its energy is limited. Therefore, it is essential to evaluate the execution cost of a manipulator when selecting a trajectory planning algorithm.

Figure 1 is our robot platform, and the application of a flexible manipulator can be broken into three levels: (a) scene perception, (b) trajectory planning, and (c) closed-loop control. Scene perception provides essential information for manipulator trajectory planning, obtaining and building a 3D model of the physical world through various sensors such as cameras and radars, and using object dictionaries and decision trees to identify all objects in a given scene. Once a three-dimensional model of the environment is created, the movement of a manipulator can be defined within this configuration space. Obstacles can also be defined, allowing the best-fitting path sequence to be obtained from a trajectory planning algorithm, which involves the kinematics of a manipulator and significant inverse kinematics. Since a manipulator usually has multiple degrees of freedom, multiple solutions for a target position may exist. A trajectory planning algorithm must be applied to obtain a series of joint values to reach a target position successfully since inevitable errors in the recognition system and manipulator movements must be constantly corrected. Closed-loop control can minimize the uncertainty associated with each operation, reducing the risk of failure and improving operational efficiency. The above (a), (b), and (c) can constitute a complete control cycle, and coordinating all levels in each iteration proves beneficial because the dynamic management of operations can prevent severe consequences due to excessive accumulated errors in continuous state updates. Although all layers must be considered and verified in robotic systems, this paper focuses on the second layer (b), which involves only the trajectory planning of a manipulator. 

## 3. Related Works

Since this paper only discusses the kinematics of a manipulator, the standard D–H method (Denavit–Hartenberg) is explained below [9]. For connecting rod *i* − 1, first, take the distal axis of connecting rod *i* − 1 (the joint axis *i*) as the z*_i_*_−1_ axis and the standard perpendicular line between the joint axis *i* − 1 and the *i*-axis as the x*_i_*_−1_ axis. The right-hand rule determines the y*_i_*_−1_ axis and establishes the coordinate system accordingly. The coordinate transformation sequence is as follows: the x*_i_*_−1_ axis rotates around the z*_i_*_−1_ axis by an angle θ*_i_*, O*_i_*_−1_ moves d*_i_* along the z*_i_*_−1_ axis, O*_i_*_−1_ moves a*_i_* along the *x*-axis, and the z*_i_*_−1_ axis rotates around the *x*-axis through an angle α*_i_*. The above transformation can convert the coordinate system O(*i* − 1) to O(*i*). The matrix in Figure 2 can represent the pose transformation between each link.

The above is the forward kinematics model of an articulated manipulator. That is, through the transformation of each joint from the base, the position and posture of the end-effector (Tool Center Point, TCP) can finally be determined. On the contrary, given the pose of a TCP, the state of each joint must be calculated, which is called an inverse kinematics solution. In addition, there is a method called the modified D–H method (Modified Denavit–Hartenberg) [10]. The coordinate system of the modified D–H method is at the proximal end of the connecting rod, and the transformation sequence is from the *X*-axis to the *Z*-axis.

Based on the standard D–H, Refs. [11,12] proposed a double A* method, which divides trajectory planning into two stages: (a) the approach stage and (b) the precision stage. First, roughly approach the target position with giant steps and then move forward more accurately with small steps. This method has significant advantages over traditional inverse kinematics pathfinding regarding execution time and the number of steps. Still, it impacts the manipulator joints because it produces too many direct trajectories. Ref. [13] explained how to generate manipulator motion trajectories with different optimal criteria. Multiple trajectory algorithms use the minimum time criterion, which may cause rapid wear of the actuator due to discontinuous motion, induce vibration, and deteriorate tracking accuracy [14]. The remedy is to improve the motion trajectory through a cost function that limits twitching and minimizes oscillatory behavior [15]. Ref. [16] is a doctoral thesis that researched trajectory planning with cost assessment for a manipulator, and it presented three methods for minimizing time cost. Refs. [17,18] discussed linking two points in an operational space while minimizing a cost function, considering dynamic equations of motion and bounds on joint positions, velocities, jerks, and torques. Ref. [19] studied two trajectory planning problems. The first case involves an end-effector that is constrained to move along a prescribed path in the robotic workspace. In contrast, the second case is one where the end-effector’s trajectory must be determined in the presence of obstacles. Both problems have been solved as optimal control, and one must find the trajectory and the actuator torques that minimize energy consumption during the motion. Ref. [20] addressed the problem of time-energy optimal control of a cable robot, with trajectory planning as the overall mission. It minimizes a cost function by considering dynamic equations of motion and bounds on joint torques. The cost function was chosen as a weighted balance of the actuators’ traveling time and mechanical energy. Although there are a variety of off-the-shelf trajectory planning algorithms, each algorithm has its focus, such as pursuing the shortest stroke or needing to consider time cost. Based on the requirements for robot performance in the agricultural field, the cost of trajectory planning must be evaluated, which makes it particularly meaningful. 

Existing manipulator solutions lack the flexibility to respond to changes autonomously, and most solutions need to be implemented modularly. One possible solution to overcome the limitations is to rely on the Robot Operating System (ROS, www.ros.org, accessed on 1 May 2024). ROS originated in 2007 as a collaboration between projects at Stanford University’s Artificial Intelligence Laboratory and the Personal Robots Program at robotics company Willow Garage, which has been powering it since 2008. ROS kinetic is a set of software libraries and tools that can be used to build modular robotics applications for the sake of its open-source drivers and advanced built-in algorithms. ROS allows the combination of simple algorithms to create modular solutions for complex problems, thereby increasing the flexibility of the overall system. MoveIt, a manipulator software module in ROS, comprises a series of mobile operation packages, including motion planning, operation control, 3D perception, kinematics, collision detection, and friendly GUI. A manipulator can take on a variety of physical forms. This paper uses the UR5 manipulator from Universal Robots as the research object. It mainly discusses kinematic issues in the trajectory planning of an articulated manipulator and creatively proposes a Joint-Cost function (JC), which involves multiple weight factors.

## 4. Proposed Method

This paper adopts the design of the Robotics Toolbox for Python (version 1.1.1, Requires: Python >=3.7), the state-of-the-art Python version of the Robotics Toolbox for MATLAB (RTB 10.4) [21]. The main features of this version include the following. (a) It provides objects to represent rotations as matrices in SO(2) and SO(3), rigid-body motions as matrices in SE(2) and SE(3), and twists in se(2) and se(3) [22]. (b) Support models are expressed using Denavit–Hartenberg (D–H) notation (standard and modified) and elementary transform sequences [23,24]. We begin with forward kinematics but deliberately avoid the commonly used D–H parameters. Instead, we approach the problem using the elementary transform sequence (ETS), an intuitive, uncomplicated, and superior method for modeling a kinematic chain [25]. ETS avoids D–H’s unnecessary complexity and frame allocation constraints and allows joints to rotate or translate around or along any axis. This paper takes a UR5 manipulator as the object (Figure 3) to study the cost calculation method. The UR5 is a manipulator composed of extruded aluminum tubes and joints. Each joint provides one degree of freedom and there is a total of six degrees (①②③④⑤⑥ in Figure 3). Waypoints are points in a robot’s workspace, and they can be determined by moving the robot to a specific location or calculated by software. The robot passes through a series of waypoints to perform its tasks. The UR5 provides various options for how the robot moves between points, such as MoveL, MoveJ, and several other motions.

Inspired by Ref. [12], whose joint effort parameter was not elaborated, to reduce energy consumption and obtain the best performance, we provided a criterion named cost function (CF), which is the total cost of a robot’s joints. To the UR5 in Figure 3, there are six joints (N=6), each with mass. $m_{i},i=1\ldots N$. The total mass driven by joint 1 is $\sum_{i=1}^{N}m_{i}$, and we mark it as $M_{1}$. Therefore, joint 2 drives total mass $M_{2}=\sum_{i=2}^{N}m_{i}$, …, till joint 6 drives $M_{6}=m_{6}$, and we obtain one tuple $\left[M_{1},M_{2},M_{3},M_{4},M_{5},M_{6}\right]$. The change in each angle can quantize the variation of joint 1, and here, we mark it as $\omega_{1}$, obtaining another tuple $\left[\omega_{1},\omega_{2},\omega_{3},\omega_{4},\omega_{5},\omega_{6}\right]$ for all joints. Now, we can formalize CF as Formula (1):(1)$$CF=\sum_{i=1}^{N}\omega_{i}*M_{i}=\sum_{i=1}^{N}\left(\omega_{i}*\sum_{j=i}^{N}m_{j}\right)$$

From UR5’s Datasheet, we cannot find the mass of each joint. For easy discussion, we set $M_{i}=N+1-i$, which assumes that each joint has the same unit mass so that we can obtain a constant mass tuple [6,5,4,3,2,1]. In fact, the essence of joint quality is to consider the minimum quality cost. When the bottom joints rotate, the high joints on them are driven to move together. Therefore, each joint’s known or unknown specific mass value has no material effect on the contribution of the cost assessment. In Section 5.4, we discuss this issue further.

Furthermore, there are always uncertain iteration steps in inverse kinematics, which means time consumption, so we added iteration (Iter) as the second factor to CF. The final position searched by inverse kinematics cannot be identical to the respective one. We called residual here for its absolute value of the difference (Resi) and added it to CF as the third factor. Now, we can describe CF further as Formula (2), and we take Formula (1) as Mass:(2)$$CF=\emptyset_{1}Mass+\emptyset_{2}Iter+\emptyset_{3}Resi=\emptyset_{1}\sum_{i=1}^{N}\left(\omega_{i}*\sum_{j=i}^{N}m_{j}\right)+\emptyset_{2}Iter+\emptyset_{3}Resi$$

For generalization, we always attempted M times (M position pairs [P_src_, P_dst_]), averaged each factor (mathematical expectation of equal probability), and summed the total to obtain Formula (3).
(3)$$CF=\emptyset_{1}\frac{\sum_{k=1}^{M}{Mass}_{k}}{M}+\emptyset_{2}\frac{\sum_{k=1}^{M}{Iter}_{k}}{M}+\emptyset_{3}\frac{\sum_{k=1}^{M}{Resi}_{k}}{M}\;=\emptyset_{1}\frac{\sum_{k=1}^{M}{\left(\sum_{i=1}^{N}\left(\omega_{i}*\sum_{j=i}^{N}m_{j}\right)\right)}_{k}}{M}+\emptyset_{2}\frac{\sum_{k=1}^{M}{Iter}_{k}}{M}+\emptyset_{3}\frac{\sum_{k=1}^{M}{Resi}_{k}}{M}$$

The magnitude order of each factor may have a huge difference, so we compared each factor to obtain a ratio and then summed the total, as shown in Figure 4.

Methods used to determine the weight coefficients include the expert evaluation method, principal component analysis method, analytic hierarchy process, entropy weight method, fuzzy comprehensive evaluation method, data mining method, etc. All three factors have different concerns: (a) Mass is related to energy consumption, (b) Iteration is related to time consumption, and (c) Residual is related to position accuracy. Usually, the coefficients of factors are related to a certain task goal, such as our robot platform, which is designed for weeding and can be equipped with various weeding terminals. Therefore, it does not make much sense to discuss the coefficients of factors in isolation from a specific operational objective. According to the task and state of a robot, we can dynamically adjust the coefficients of factors to budget the cost and choose different inverse kinematics methods to generate a desired trajectory sequence. If the robot is low in power, the coefficient of Mass will be increased, while the coefficient of Iteration will be increased for speed and the coefficient of Residual will be increased for high precision. It is possible to collect the task parameters and the status of the robot, and a program will automatically determine and adjust the coefficients of factors, which is currently underway. For easy discussion, we set $\left[\emptyset_{1},\emptyset_{2},\emptyset_{3}\right]$ to [1, 1, 2], which means that mass and iteration have the same weight, while residual causes more concern for the sake of precision.

Now, we can calculate every inverse kinematics configuration to obtain CFs and find the minimum one. Firstly, we list algorithms to be evaluated, including three algorithms, as follows: (a) Newton–Raphson (NR) [26], (b) Gauss–Newton (GN) [27], and (c) Levenberg–Marquardt (LM) [28] with variants: Wampler’s Method, Chan’s Method, and Sugihara’s Method. The overall structure of our work is shown in Figure 5, and our cost calculation method can be described as Algorithm 1.
**Algorithm 1:** Cost Calculation for Manipulator Trajectory PlanningInput: position pair [P_src_, P_dst_]              IK methods list: [NR, GN, LM, …]              UR5 joints’ config: N = 6, M1,M2,M3,M4,M5,M6              Influence factors of [Mass, Iter, Resi]: [∅1,∅2,∅3]Output: costs (CF)Let IK = [NR, GN, LM, …]Let [G, I, R] = [0, 0, 0]// Run each IK method to search trajectoriesK = size of IKfor i = 0, …, K           Get IK_i_           [J_i_, I_i_, R_i_] = IK_i_ ([P_src_, P_dst_]), where J_i_ = [ω1,…,ωN]           G_i_ = 0           for j = 0, …, N                      Gi+=ωj∗Mj           end for           Get [G_i_, I_i_, R_i_] and save           [G, I, R] += [G_i_, I_i_, R_i_]end for// Finally, calculate costsCF = [0,…,0]for i = 0,…,K           CFi=∅1 × G_i_/G + ∅2 × I_i_/I + ∅3 × R_i_/R end forreturn CF

Our main contributions include the following: (a)Instead of the conventional D–H method, we utilized the state-of-the-art Python version of the Robotics Toolbox for manipulator trajectory planning.(b)We proposed a cost function with mass, iteration, and residual to assess the effort of a manipulator. The arm actuation cost, the time cost of trajectory steps, and the operation accuracy are comprehensively considered.(c)We realized three inverse kinematics methods with variants: NR, GN, and LM. We verified our cost function’s feasibility and effectiveness. Furthermore, we compared our methods with the state-of-the-art techniques, Double A* and MoveIt.

In the next section, we design three experiments as follows:Exp1:CF calculation based on the UR5’s ETS and comparisons between NR, GN, and LM with variants.Exp2:Comparisons with the state-of-the-art methods, Double A* and MoveItExp3:Winning method deployed on the UR5.Exp4:Mobile operation.

## 5. Experiment and Discussion

### 5.1. CF Calculation Based on the UR5’s ETS and Comparisons between NR, GN, and LM with Variants

Firstly, we used ETS to describe the UR5 and randomly generate ten position pairs [P_src_, P_dst_]. The NR and LM Chan methods were used to search trajectories. Figure 6 shows a UR5 simulation in Swift and plot [Mass, Iter, Resi] values.

Table 1 normalizes the average [Mass, Iter, Resi] values. LM (Chan λ = 0.1) obtains the min CF value for its [Mass, Iter], which is more minor than NR (pinv = False), especially [Mass]. We confirm that the joints’ effort of LM (Chan λ = 0.1) is graceful.

Similarly, Table 2 normalizes the average [Mass, Iter, Resi]. Here, we performed tests on NR, GN, and LM with variants. Although the mins of [Mass, Iter, Resi] occur in different methods, LM (Chan λ = 0.1) still obtains the best CF value.

### 5.2. Comparisons with the State-of-the-Art Methods, Double A* and MoveIt

Experimental validation in Ref. [11] compared Double A* and MoveIt, and we adopted the same parameters as input. The position pair [P_src_, P_dst_] is [(0.126, 0.095, 0.466), (0.100, 0.075, 0.725)], and Figure 7 plots the trajectories. We noticed that both Double-A* and MoveIt searched surrounding areas to find relatively good paths, while NR, GN, and LM with variants explored paths in leaps. LM with variants displayed less fluctuation than the others. 

Table 3 shows that the steps of Double A* and MoveIt are more than our inverse kinematics methods, but the distances are in-between ours. Notice that the CF of MoveIt cannot be calculated for the sake of joints not being given. Still, we obtained the best CF by LM (Chan λ = 0.1). As for execution time, it does not make much sense to run the experiments on different computers.

### 5.3. Winning Method Deployed on the UR5

Figure 8 shows the trajectories of a UR5 generated by LM (Chan λ = 0.1). We can find that the motions of the first two are huge, while the others are tiny. The UR5 realizes trajectory motion using waypoints, so there are six waypoints. We chose MoveL, a movement type on the UR5 from one point to another, to keep TCP motion in a line. 

### 5.4. Mobile Operation

As shown in Figure 1, our robot platform installed Husky software (clearpath-universal-kinetic-amd64-0.4.17) and ROS kinetic on the vehicle’s desktop computer. Start the onboard desktop computer (other devices of the platform have been connected via USB or network cable), start Husky (the communication indicator is solid green to indicate that ROS communication is regular), and start UR5 (the teach pendant is connected).

Open a terminal window and start roscore with the following command:

roscore

Start the depth camera to obtain the depth information of the target, and the command is as follows:

roslaunch realsense2_camera rs_rgbd.launch

Start the UR5 mobile operation program (Figure 9 and Figure 10) with the following commands:

roslaunch ur_pick ur_pick.launch

We obtained a position pair [P_src_, P_dst_]: [(0.373, −0.108, 0.383), (0.747, 0.004, −0.193)], and the inverse kinematics solution process of LM (Chan λ = 0.1) is as follows:

a new search … 

start joint is 

[0.01273444 −1.45818111 −1.24780111 −0.24265222 1.54784556 3.22495444]

iter joint is 

[0.09502918 −1.80106334 −1.93388123 −0.75895702 1.59453023 3.12889729]

…

iter joint is 

[0.15525559 −2.25504331 −1.24797565 −1.08260196 1.54767081 3.22512886]

Find good iter joint!

LM (Chan λ = 0.1) joint effort is 26.76, pos:search 1:1, iter = 5, E = 7.98

LM (Chan λ = 0.1) END-success

LM (Chan λ = 0.1), time cost: 0.08 s

Now, for the issue of UR5’s unknown joint mass in Section 4, we assumed $m_{1\ldots 6}=\left[6,5,4,3,2,1\right]$, not consistent [1,1,1,1,1,1] as before, then we obtained $M_{1\ldots 6}=\left[21,15,10,6,3,1\right]$. After inverse kinematics searching, we obtained a total joint effort of 63.93, which is bigger than before but still the minimum!

## 6. Conclusions

Cost calculation to select a suitable trajectory planning algorithm for a manipulator is meaningful, especially for an agricultural robot. This paper uses the state-of-the-art Python version of the Robotics Toolbox for manipulator trajectory planning. It is more intuitive and easy to use than the conventional D–H method. We proposed a cost function with mass, iteration, and residual to calculate the total effort of a manipulator. Several factors are considered more comprehensively than existing methods. We realized three inverse kinematics methods (NR, GN, and LM with variants) and verified our cost function’s feasibility and effectiveness. To further ascertain the generality of our process, we compared it with the state-of-the-art techniques, Double A* and MoveIt. Experiments have shown that LM (Chan λ = 0.1) always obtained the best result, which confirms that our method is valid and stable. Finally, LM (Chan λ = 0.1) is applied in mobile operation on our agricultural robot platform.

Although the kinematics approach is straightforward, it experiences some problems in implementation because of a lack of inertia and torque constraints. For example, algorithms that find the optimal trajectory of a manipulator in obstacle avoidance are essential for better results. Thus, this is a meaningful direction. The performance index of a manipulator seeks to quantify the performance under a given configuration. The maneuverability index and the condition number are two common performance indicators. Further research directions of this paper can include these performance indicators in the evaluation system.

## Figures and Tables

**Figure 1 sensors-24-04096-f001:**
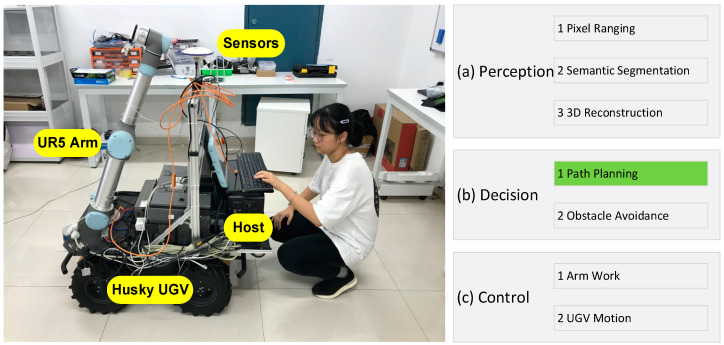
Our robot platform. (**Left**): the robot entity. (**Right**): a three-tier architecture. The green label is our focus.

**Figure 2 sensors-24-04096-f002:**
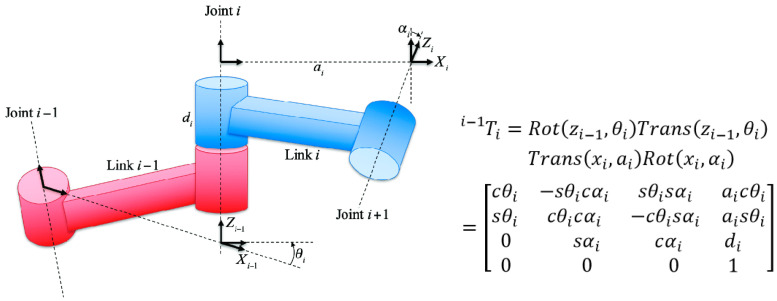
(**Left**) structure of the standard D–H method. (**Right**) link pose transformation matrix of an articulated manipulator.

**Figure 3 sensors-24-04096-f003:**
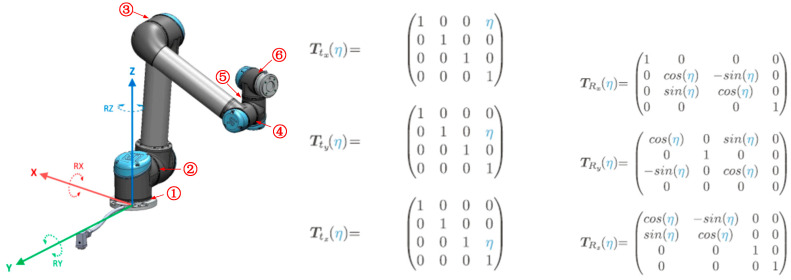
(**Left**) a UR5 with six freedoms. (**Right**) homogeneous transformation matrixes representing translations and rotations. M elementary transforms, for the UR5, M = 13, and the ETS is SE3(0, 0, 0.08916)⊕Rz(q0)⊕SE3(0, 0.1358, 0; 0°, 90°, −0°)⊕Ry(q1)⊕SE3(0, −0.1197, 0.425)⊕Ry(q2)⊕SE3(0, 0, 0.3922; 0°, 90°, −0°)⊕Ry(q3)⊕SE3(0, 0.093, 0)⊕Rz(q4)⊕SE3(0, 0, 0.09465)⊕Ry(q5)⊕SE3(0, 0.0823, 0; 0°, −0°, 90°).

**Figure 4 sensors-24-04096-f004:**
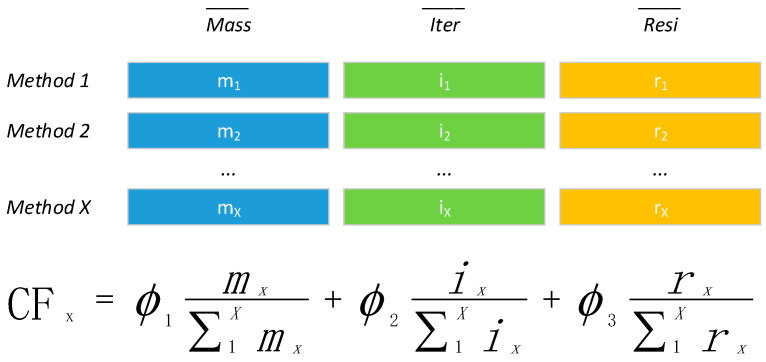
CFx calculation diagram. (**Top**) X methods with averages of mass, iteration, and residual. (**Bottom**) CFx formula.

**Figure 5 sensors-24-04096-f005:**
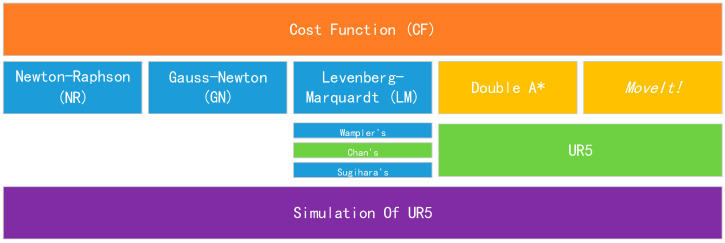
The overall structure of our work. (**Top**) cost function we proposed. (**Middle**) NR, GN, and LM with variants to be tested; Double A* and MoveIt from Ref. [11]; LM Chan’s method deployed on the UR5. (**Bottom**) All processes are simulated on the UR5.

**Figure 6 sensors-24-04096-f006:**
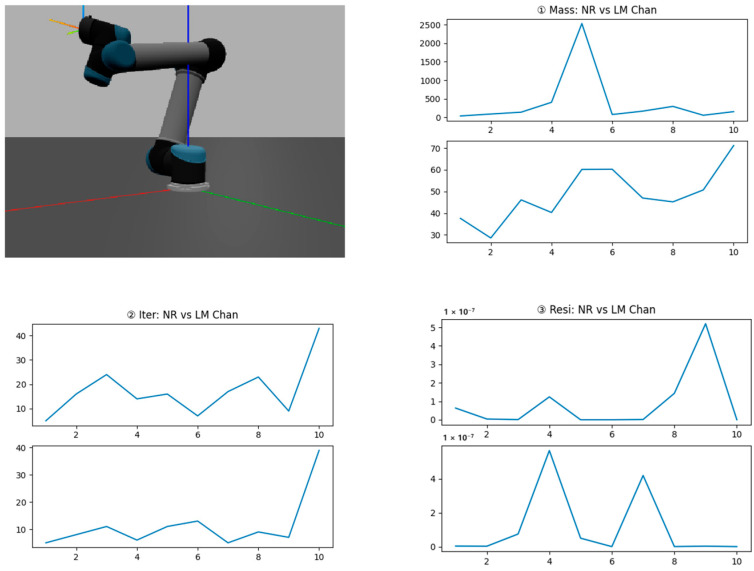
Comparison of the NR and LM Chan methods. (**Top left**) a UR5 visualized in Swift, and red, green, blue lines stand for coordinates X, Y, Z respectively. (**Top right**) Mass curve. (**Bottom left**) Iter curve. (**Bottom right**) Resi curve.

**Figure 7 sensors-24-04096-f007:**
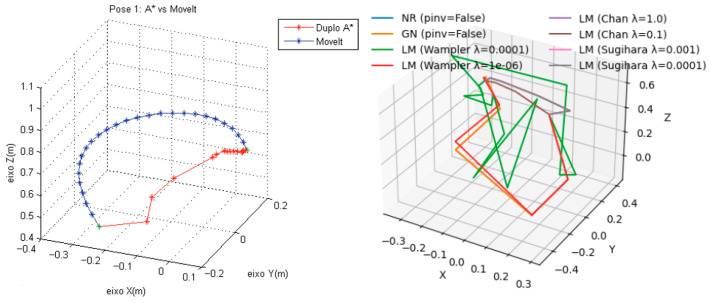
(**Left**) trajectories of Double A* and MoveIt. (**Right**) trajectories of NR, GN, and LM with variants.

**Figure 8 sensors-24-04096-f008:**
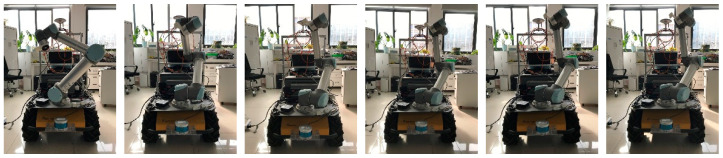
From left to the right is the sequence of a UR5.

**Figure 9 sensors-24-04096-f009:**
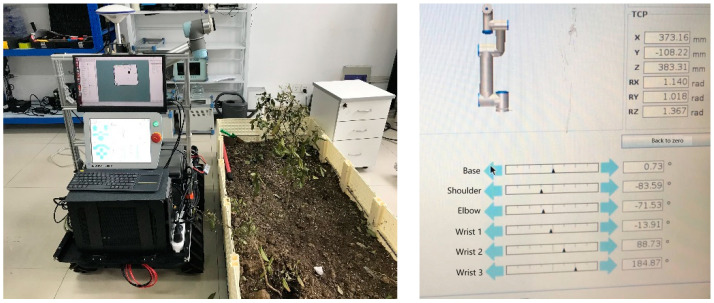
Initial posture of the manipulator.

**Figure 10 sensors-24-04096-f010:**
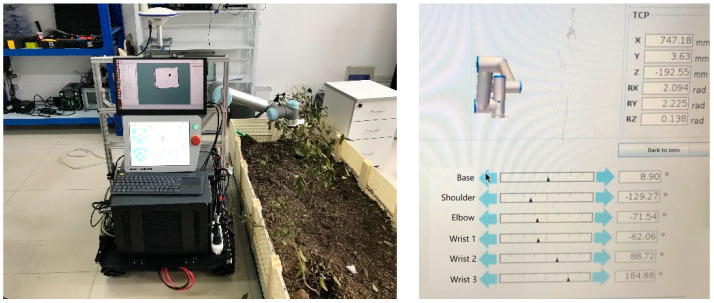
Manipulator working posture.

**Table 1 sensors-24-04096-t001:** Comparison between NR and LM over ten iterations.

Method	$\overset{\_}{Mass}$/Ratio	$\overset{\_}{Iter}$/Ratio	$\overset{\_}{Resi}$/Ratio	CF
NR (pinv = False)	395.63/0.89	17.40/0.60	8.57 × 10^−8^/0.43	2.35
LM (Chan λ = 0.1)	48.73/0.11	11.40/0.40	1.12 × 10^−7^/0.57	1.65

**Table 2 sensors-24-04096-t002:** Comparison between NR, GN, and LM with variants over ten iterations.

Method	$\overset{\_}{Mass}$/Ratio	$\overset{\_}{Iter}$/Ratio	$\overset{\_}{Resi}$/Ratio	CF
NR (pinv = False)	395.63/0.13	17.40/0.10	8.57 × 10^−8^/0.06	0.35
GN (pinv = False)	1619.35/0.55	27.80/0.15	8.03 × 10^−8^/0.05	0.81
LM (Wampler λ = 0.0001)	142.17/0.05	19.40/0.11	2.55 × 10^−7^/0.17	0.50
LM (Wampler λ = 1 × 10^−6^)	585.76/0.20	38.80/0.21	1.40 × 10^−7^/0.09	0.60
LM (Chan λ = 1.0)	48.10/0.02	18.60/0.10	1.27 × 10^−7^/0.09	0.29
LM (Chan λ = 0.1)	48.73/0.02	11.40/0.06	1.12 × 10^−7^/0.08	0.23
LM (Sugihara λ = 0.001)	43.73/0.01	28.00/0.15	4.57 × 10^−7^/0.31	0.79
LM (Sugihara λ = 0.0001)	48.15/0.02	19.30/0.11	2.23 × 10^−7^/0.15	0.43

**Table 3 sensors-24-04096-t003:** Comparison between MoveIt, Double A*, and our inverse kinematics methods.

Method	Steps (Iter)	Distance (m)	Execution Time (ms)	CF
Double A*	20	0.81	4277	1.13
MoveIt	28	1.13	5894	—
NR (pinv = False)	9	2.21	145	0.44
GN (pinv = False)	9	2.20	140	0.44
LM (Wampler λ = 0.0001)	17	5.55	263	0.96
LM (Wampler λ = 1 × 10^−6^)	8	2.16	124	0.42
LM (Chan λ = 1.0)	7	0.57	108	0.16
LM (Chan λ = 0.1)	5	0.43	78	0.12
LM (Sugihara λ = 0.001)	7	0.59	109	0.17
LM (Sugihara λ = 0.0001)	7	0.58	107	0.16

## Data Availability

The experimental data will be made available by the authors.

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
