# Peer review of "A Novel Cost Calculation Method for Manipulator Trajectory Planning"

_sensors, 2024, doi:10.3390/s24134096_

Round 1
Reviewer 1 Report
Comments and Suggestions for Authors
In order to improve the energy-saving effect during the operation of robots, this article proposes a cost function with mass, iteration, and residual to calculate the total effort of a manipulator. The cost function's feasibility and effectiveness were verified through experiments, and compared with other algorithms, which confirms that the method is valid and stable. Although the issues studied in the paper are relatively important, the innovation in the content of the paper is not prominent enough, and there are still the following problems:
1. The literature review on trajectory planning is not comprehensive enough, and further summary and induction of the current research status are needed.
2. How was Cost Function established? Why not consider joint acceleration, end load, etc? The establishment of the existing Cost Function is unreasonable.
3. In the Cost Function calculation, it is unreasonable to assume that the mass of each joint of the UR5 robot is consistent.
4. What are the criteria for selecting the three weight coefficients? The paper did not explain clearly.
5. What is the specific task of the final experiment? What is the task objective? It is not possible to affect specific task objective in order to reduce Cost Function.
Author Response
Thanks for your comments. Please download R1.pdf

Reviewer 2 Report
Comments and Suggestions for Authors
This study proposes a cost calculation method for manipulator trajectory planning. The topic is interesting, but the manuscript is not well-written and well-prepared.
I have some comments and suggestions regarding the proposed study and manuscript:
1. The author appears to have included content crossly on background, related work, and research motivation in the first two sections, causing confusion for the readers. For instance, some content regarding the background of the research was included in the 'Related Work' section, while the 'Introduction' contained a review of the related work. It is recommended that these two sections be divided into three parts: 'Introduction,' 'Background,' and 'Related Work.' Each part should then focus on providing a detailed explanation of the respective content.
2. The methods for comparison, Double A* and MoveIt!, were proposed more than ten years ago. It is doubtful that these two methods still belong to those SOTA techniques. Please explain the rationale for selecting such old methods as the compared methods. Can you include some recently proposed methods in the experiment?
3. The academic quality of the English expression throughout the entire article is insufficient. It is advisable to thoroughly read the entire manuscript and make careful revisions.
Comments on the Quality of English LanguageThe academic quality of the English expression throughout the entire article is insufficient. It is advisable to thoroughly read the entire manuscript and make careful revisions.
Author Response
Thanks for your comments. Please download R2.pdf

Reviewer 3 Report
Comments and Suggestions for Authors
**Reviewer Comments on Manuscript ID sensors-3004689: "A Novel Cost Calculation Method for Manipulator Trajectory Planning"**
**General Comments:**
The manuscript presents a study of Cost Calculation Method for Manipulator Trajectory Planning, focusing on cost function with mass, iteration, and residual to assess the effort of a manipulator
. While the paper contributes to the review of new planning rout, several major revisions are required to strengthen the manuscript's contribution further.
**Major Concerns:**
1. **Novelty issues:**
- The author summarized three main contributions: use ETS instead DH is not real contribution, it is more like author choose another simpler method; the third point is just verification. Only the cost function with mass, iteration, and residual can be consider as novel contribution, but the author didn’t deeply dig this point.
2. **Design rationality issues:**
- For the cost function, the coefficient setting of the three items or how to combine the three cost is key part, and the author just choose 11 2 for no reason, which make the methodology extremely imprecise.
3. **Design analysis for application goals :**
-At the begging of the article, the author mentioned about application of agriculture, but all the work of the article is nothing special for agriculture, no constraint condition has been list out.
4. **The main content generally does not require coding :**
The code between 223-224 is simple and illustrate nothing, it is not required.
Author Response
Thanks for your comments. Please download R3.pdf

Round 2
Reviewer 1 Report
Comments and Suggestions for Authors
Although the authors have answered all the questions, there are still some questions that need further clarification.
1. Just because you don't know the mass of each joint of the UR5 robot, you can't say that the mass of the joints has no impact on the research. We need to add simulation analysis to prove that the mass of each joint has no impact on this study.
2. For the final experiment, the initial trajectory should be given first, and then the trajectory optimized by considering Cost Function should be given. Compare these two trajectories to analyze whether they can meet the experimental indicators and save energy.
Author Response
Please download R1-2.pdf, thanks!

Reviewer 2 Report
Comments and Suggestions for Authors
I have no more questions.
Comments on the Quality of English LanguageThe Quality of English Language still can be improved.
Author Response
Please download R2-2.pdf, thanks!

Reviewer 3 Report
Comments and Suggestions for Authors
"Weight coefficients are really hard to determine" is not a reasonable reply, without the improvement of this part, the scientific soundness of the article is short.
Author Response
Please download R3-2.pdf, thanks!

Round 3
Reviewer 3 Report
Comments and Suggestions for Authors
no more